# Response of UK Horse, Pony and Donkey Owners to the Early Stages of the COVID-19 Pandemic

**DOI:** 10.3390/ani11051215

**Published:** 2021-04-23

**Authors:** Jo Hockenhull, Catherine Bell, Jo White, Suzanne Rogers

**Affiliations:** 1Human Behaviour Change for Animals (HBCA), Norfolk NR9 4DE, UK; 2Equine Behaviour and Welfare Association, Surrey GU8 6AX, UK

**Keywords:** COVID-19, equine welfare, human–animal relationships

## Abstract

**Simple Summary:**

COVID-19 spread across the UK in early 2020. On 16 March 2020, the government announced measures that could be taken by the public to reduce the spread of the virus. On the 18 March 2020 schools across the UK were closed and by 23rd March 2020, the UK entered its first mandatory lockdown. During this time, UK equestrians had to balance meeting their horse’s welfare needs with government advice and guidance. A survey was distributed to equine owners at the end of March 2020 via social media to try to understand their equine-related decision-making at this time. Data from 452 respondents indicated differences between the types of changes made following the government announcements on 16 March and 18 March. Early changes were focused on implementing biosecurity measures at equine establishments and opting to stop riding the horse, while those after 18 March were largely aimed at reducing the risk associated with various equine-related activities. The findings suggest that few equestrians had emergency plans in place should they be unable to care for their horses, donkeys or mules, and also raise questions about risk perception.

**Abstract:**

In the UK, March 2020 was a time of great uncertainty as COVID-19 became increasingly widespread. The government responded by making suggestions about how people could reduce the risk of spread on 16 March, moved swiftly into closing schools on the 18 March before announcing a mandatory lockdown on the 23rd March. This was a challenging time for UK equestrians who had to balance maintaining their equine’s routine and daily care alongside the increasing biosecurity measures. A cross-sectional survey was distributed to UK equestrians via social media over two days (28 and 29 March 2020) to better understand the decisions made by UK horse, pony and donkey owners during this time. Data from 452 respondents were generated across all four countries comprising the UK, although there were no significant differences in owner response to the pandemic between locations. The changes respondents made differed between the 16th and the 18th of March 2020, with an early emphasis on improving yard biosecurity and opting to stop riding, as well as reducing the time spent at the yard. After the 18 March, respondents placed more emphasis on risk reduction by changing the activities they did with their horse, including riding, with common examples including avoiding “high risk” activities such as riding on busy roads, jumping, riding young or nervous horses. Few respondents reported having an emergency plan in place should they become ill or otherwise unable to care for their equine. The findings highlight areas that would significantly benefit from in-depth investigation in future research. Equestrian behaviour and mindset around risk-taking and risk perception have already been researched in relation to equestrian activities and sport but have received little attention in the context of wider health challenges. Understanding the uptake of emergency planning and preparation in the UK equestrian community also warrants consideration. Using this information effectively to promote forward planning is likely to be of great benefit in equestrian responses to future health or climate-related crises.

## 1. Introduction

The early days of the COVID-19 pandemic brought times of great uncertainty [1], as well as rapid changes in the advice and guidance received from government and professional bodies as the virus spread. In the UK, life remained much the same as it had previously until late February 2020. As the virus swept across the UK, the public began to receive guidance from the government detailing measures individuals could take to reduce spread (see Figure 1 for a timeline of key events in the UK January to March 2020 relating to COVID-19). On the 23rd March, the UK government announced a mandatory lockdown where the population was informed they must remain at home unless exercising or undertaking essential work or travel [2,3,4]. This was the first time in modern history that the UK population had been subjected to such extreme measures to control a human disease outbreak, and understandably the restrictions adversely impacted everyday life and mental wellbeing [1,3,5,6,7].

While animal ownership or caregiving responsibilities might assuage some of the detrimental impacts of the current crisis, the pandemic is likely to escalate existing financial and practical concerns associated with owning or caring for an animal [5,8,9,10]. The impact of the pandemic on the practicalities of providing care for horses, donkeys and mules in the UK forms the focus of this study. Officially, visiting your horse for their essential care was one of the exclusions to the call to stay at home [11]. Despite this, rumours and mixed messages abounded during this time and many UK equestrians lacked clarity on what was allowed within the restrictions imposed by the government [4]. Equestrian bodies, including The British Horse Society, made recommendations that people should stop riding to reduce the risk of accidents, which would place an extra burden on the National Health Service at a time when demand for their services was already high [11], but this was not mandatory.

The response of animal owners and caregivers to the COVID-19 pandemic has been investigated in several survey-based studies, which have typically focused on the pros and cons of companion animal ownership at this time, alongside the impact this has on human physical and mental health [1,8,10]. Equines fall under the British Small Animal Veterinary Association (BSAVA) definition of companion animal [12] and most equestrians consider their horses this way and have strong feelings of attachment towards them [13,14]. Yet horses have been largely overlooked in the published literature regarding the impact of COVID-19 on pets and their owners, which has tended to focus on cats and dogs (e.g., [1]). As companion animals that typically do not reside with their owner or caregiver, equines are distinctly different from other companion species and this might be reflected in how their owners and caregivers responded to the pandemic with respect to their equine-related activities.

This focused cross-sectional survey aimed to better understand the decisions made by UK horse, pony and donkey owners during the early stages of the COVID-19 pandemic in March 2020. 

## 2. Materials and Methods 

### 2.1. Survey Design

A short online cross-sectional survey consisting of sixteen questions (ten closed questions, including one tick-all-that-apply question, and six open-ended questions) was created using the Google Forms facility in Google Docs. The survey was designed to be quick and easy for respondents to complete given the heightened stress and uncertainty associated with this time, and was launched five days after the first UK lockdown began on 23 March 2020. The timing was intentional to minimise any recall bias that may arise if the survey was launched later on given the speed of change experienced in the UK during March 2020. 

The opening page of the survey provided information to the participants, including who developed the survey and their contact details, the purpose of the research, the estimated time to complete the survey (five minutes) and assured potential participants of the anonymity of their responses. Interested respondents then proceeded to the next page, which detailed the survey terms and finished with statements that the respondent was taken as consenting to by actively choosing to proceed with the survey. The first page of the survey properly requested demographic information on respondents’ age, occupation, country of residence within the UK, their risk status in relation to their vulnerability to COVID-19 infection, and whether they had ever received hospital treatment for an equine-related accident. Respondents were then asked ten questions about their equine-related activities including their level of equine contact prior to March 2020 and whether they had made changes to this or stopped riding as a result of the COVID-19 restrictions announced at two time points. These were (i) the restrictions suggested by the UK government on 16 March 2020 to stop unnecessary travel and social contact, and to work from home if possible—these were suggestions rather than mandatory; and (ii) following the closure of schools on 18 March 2020 [2].

Prior to going live, the online survey and data collection protocol received ethical approval via the Equine Behaviour and Training Association (EBTA) Ethical Review process (reference number 002). EBTA follows guidance from The British Psychological Society (BPS) as outlined in their Code of Ethics and Conduct which is available online [15]. Accordingly, participants were given an introductory briefing stating the purpose and content of the study. They were informed of their rights to anonymity and data protection, including data storage and disposal following the end of the study, as well as their freedom to withdraw at any point. Participants were subsequently asked for their informed consent via a “forced choice” response. Given the timing of the survey, it was recognised in the introductory material that it was a stressful time for everyone and that the survey was optional. Finally, a debriefing was provided following the completion of the survey.

### 2.2. Subjects

The survey was restricted to people in the UK aged 18 years and over who usually have contact with horses, ponies and donkeys, through their recreational activities or employment within this sector.

### 2.3. Recruitment Activity

The survey went live at 17:00 on 28 March 2020. Due to the speed at which official guidance was changing at this time, participants were asked to complete the survey before midnight on 29 March 2020. Subjects were recruited via Facebook posts and emails from the research team and their organisations (EBTA and Human Behaviour Change for Animals (HBCA)). Details of the survey were shared by third parties from these initial recruitment activities (pass-along effect [16]). 

### 2.4. Analysis

Survey data were assembled into a Microsoft Office Excel workbook and screened prior to analysis to remove any data that did not meet the survey criteria (e.g., the respondent did not reside in the UK). Categorical variables were given numerical codes and copied into IBM SPSS Statistics (version 24 for Windows, Armonk, NY, USA) to facilitate analysis. Responses to the open-ended questions were categorised according to common themes in Excel.

Chi-square tests of independence were used to identify relationships between categorical variables. To reduce the risk of Type I error from multiple comparisons, a Bonferroni adjustment was used, resulting in an adjusted alpha of 0.0014 (0.05/35). 

Mann–Whitney U and Kruskall–Wallis tests were used as appropriate to explore the relationship between respondent age, country of residence, their COVID-19 risk status, whether they worked with equines, and whether they had previously received hospital treatment for an equine-related accident, and their reported likelihood to change their equine-related behaviour as the pandemic continues (as rated on a 5-point Likert scale).

## 3. Results

The survey was completed by a total of 467 respondents. Thirteen respondents did not meet the survey criteria (i.e., did not reside in the UK) and as a result, their data were removed from the dataset. A further two respondents completed the survey when it was accidentally reactivated between May and July. Due to concerns over recall bias, the responses from these two respondents were also discarded leaving responses from 452 respondents who completed the survey on the 28 March (30.3%) and 29 March (69.7%) 2020.

The majority of respondents were from England (*n* = 331, 73.2%; Table 1). The median age of respondents was 41–50 years and 22.3% reported that they had a high COVID risk status. Over half of respondents (*n* = 257, 56.9%) had previously received hospital treatment for an equine-related accident. Only 68 respondents (15.0%) earned their living working with equines.

Prior to March 2020, the majority of respondents had equine contact “most days” or more regularly (2 or more visits/day *n* = 54, 11.9%; everyday *n* = 18, 4.0%; most days *n* = 365, 80.8%; some point most weeks *n* = 15, 3.3%, 2 or more times a month *n* = 1, 0.2%). Respondents were asked what activities they would usually do with their equines between the months of March and September. Most respondents undertook grooming and general care (*n* = 435, 96.2%), 79.6% (*n* = 360) did the groundwork and 83.6% (*n* = 378) of respondents would usually ride their equines during these months.

Table 2 shows how respondents reported altering their equine-related behaviour following the COVID-19 restrictions announced on 16 March 2020 and the closure of schools on 18 March 2020, with the types of changes described by respondents summarised in Table 3. 

The changes respondents made differed between the 16th and the 18th of March 2020, with an early emphasis on improving yard biosecurity and stopping riding, as well as reducing the time spent at the yard. After the 18 March, respondents placed more emphasis on risk reduction by changing the activities they did with their horse, including riding, with common examples including avoiding “high risk” activities such as riding on busy roads, jumping, riding young or nervous horses. Respondents also mentioned starting to wear hats and/or body protectors while they were handling or working with their horse from the ground when they had not previously.

Respondents had a mixed response when asked to rate the likelihood that they would change their equine-related behaviour due to their perception of risk of accident or injury as the pandemic progressed. Likelihood of change was rated on a 1–5 Likert scale, running from 1 “not at all likely” to 5 “very likely”. The median value scored was 3, with 21.9% (*n* = 99) scoring 1, 13.5% (*n* = 61) scoring 2, 20.1% (*n* = 91) scoring 3, 19.2% (*n* = 87) scoring 4 and 25.2% (*n* = 114) scoring 5.

There was a significant association between respondent age and their perceived risk of contracting COVID-19, with more respondents than expected in the 51–65 and 66–75 age groups reporting that they were at high risk (χ^2^_8_ = 34.659, *n* = 451, *p* ≤ 0.001). More respondents who were at high risk of contracting COVID-19 had previously been treated in hospital for an equine-related accident or injury than was expected (χ^2^_2_ = 14.945, *n* = 449, *p* = 0.001).

Significant associations were also found between respondent risk of COVID-19 and changes to their equine-related activities after 16 March 2020 (χ^2^_2_ = 17.600, *n* = 450, *p* ≤ 0.001) and their decision to stop riding/change their training activities after 16 March (χ^2^_6_ = 25.676, *n* = 451, *p* ≤ 0.001). More high-risk respondents changed their behaviour than expected regarding their equine-related activities and decided to stop riding/change their training activities. The remainder of the analysis did not yield any significant results. 

## 4. Discussion

The COVID-19 pandemic is being increasingly viewed through the lens of the One Welfare framework, given the impact it has had on humans, animals and the environment alike, and how it has highlighted the interconnectedness between them [17,18]. The changes experienced for equines and their owners during this time represent only a small part of this whole but demonstrate how a global health challenge faced by the human population can significantly impact the lives, and potentially the welfare, of one type of domestic animal at a local level.

The findings from this survey provide an insight into the thought processes and priorities of UK equestrians during a short, but very uncertain, period of time. The early days of the COVID-19 pandemic in the UK were fraught with rapidly changing advice and guidance before the government ultimately took the step of putting the entire country into lockdown [2]. This survey captured the response of equestrians to this evolving situation within 5–6 days of the lockdown being announced on 23rd March 2020, minimising recall bias and allowing a unique dataset to be generated.

During the 35 hours, the survey was live, 452 complete responses were collected that met the survey criteria. Respondents came from all four countries comprising the UK, although it must be noted that the majority were from England and only one respondent was from Northern Ireland, perhaps due to the low numbers of Northern Irish Facebook followers of EBTA and HBCA. At this stage in the pandemic, the devolved countries of the UK were still taking the lead from decisions made by the UK government at Westminster, so the impact of this skewed distribution is minimal, as was demonstrated in not finding any significant difference in responses between respondents from different UK countries. Unusually for an online survey, there was an even distribution of respondents across the age groups, although there were no respondents aged 76 years or older, a trend observed in another online survey relating to people and their animals during the pandemic [1]. 

Under a quarter of respondents considered themselves at high risk of contracting COVID-19, and it is not surprising that significantly more of these respondents made the decision to change their equine-related behaviour and stop riding/change their training activities earlier than those who considered themselves not to be at high risk. While all humans are susceptible to COVID-19 [19], the finding that more of the respondents considered at high risk of contracting COVID-19 fell into the higher age groups (51–65 and 66–75 years old) is also consistent with the accepted the higher risk of this population. While many of the risk factors associated with COVID-19 were not well understood in the early days of the pandemic, it was recognised relatively early on that older people were at greater risk from COVID-19, and certainly by the time this survey was conducted (see [20]). More difficult to explain is that more high-risk respondents had previously received hospital treatment for an equine-related accident. This could potentially reflect the compromised health status of some members of this sample or health-related paranoia. However, it is more likely that the higher median age of this sub-sample means that they have potentially been around equines for longer and may therefore be expected to have accumulated more equine-related injuries than younger respondents (who are typically at lower risk of contracting COVID-19). The culture of UK equestrianism where taking risks is normalised may also contribute to this finding [21]. 

A greater percentage of respondents carried on their normal equine-related activities, including riding/training, after the suggested restrictions were proposed on 16 March 2020, than the closure of schools on 18 March 2020. This is consistent with the messaging equestrians received at these times from UK equine bodies [11], although it contrasts with the reduction in equestrian activities reported in the USA following the World Health Organisation (WHO) designating the COVID-19 outbreak a pandemic on 11 March 2020 [2,22,23]. Reports of equestrians feeling socially pressured not to ride were not uncommon [4], however it appears that many continued to ride their horse despite this. Understanding this mindset is an important subject for future research.

After the 16 March 2020, respondents reported the introduction of biosecurity measures at the yards where their horses were kept. The rigor of these measures differed between respondent reports from simply abiding by social distancing regulations while on the yard, to disinfecting all surfaces touched, using disposable gloves, and ensuring that the respondent was the only person touching their horse and all their associated equipment. These measures closely resemble those used in equine establishments in Ontario [13]. Facemasks were not reported by respondents as one of the biosecurity measures they used while with their horses. A study of equestrian establishments in Ontario similarly found that facemasks were not used by the majority of people [13]. 

After 16 March 2020, respondents also reported changes in the amount of time they spent at the yard and how often they visited in comparison to their routine pre-March 2020, as found by [4]. Some respondents mentioned the establishments of rotas detailing who could visit the yard and when, while others reported that they stopped visiting their horses altogether either on their own initiative or at the request of the yard owner.

Following the closure of schools on 18 March, more respondents appeared to focus on minimising the risk involved in their equine-related activities. This included riding in areas perceived to be less risky such as arenas and on quiet lanes, no longer participating in jumping (in the school or outside), no longer taking their horse for walks in-hand, and putting off plans to back young horses or bring others back into work. Equestrians typically accept that there is always an element of risk involved in interacting with equines [24], and it should be noted here that those respondents who did not report taking actions to minimise risk in response to the announcements on 16th and 18 March 2020 may have already taken steps to reduce the risks associated with their equestrian activities well before the pandemic. 

Riding on roads is known to be a high-risk activity, with over 60% of UK riders reporting to have experienced a near miss or accident when riding in this environment [25]. Data from a South Australian hospital demonstrated a reduction in equestrian-related traumatic injuries received during 2020 when COVID-19 restrictions were in place, when compared with the same period in 2019, suggesting that Australian equestrians had also taken measures to reduce their risk of injury [26]. A UK-based study reported a reduction in sports-related injuries during lockdown but did not specifically mention equestrianism [27]. Respondents also said they had stopped travelling their horses to other locations for riding lessons or competitions, and riding instructors were no longer allowed to come to their premises to give them lessons, both of these changes are likely to reduce the risk of accidental injury in multiple ways. Some respondents (<10%) at both time points mentioned changing how their horse was managed due to the pandemic, including switching their horse onto full livery (where the yard owner or staff would take full responsibility for their daily care), roughing horses off (by turning them out into fields, typically without rugs to reduce the work involved in mucking out and changing rugs for themselves, or if a third party had to take over caring responsibilities), even moving their horses to more private premises to reduce the risk associated with being on a busy yard. This is in contrast to Williams et al. [4], who found that the majority of UK horse owners they surveyed reported changes in their horse’s management due to COVID-19, typically due to the requirements of their yard owner rather than their personal choice. 

Very few respondents reported taking measures to prepare for potential challenges ahead. For example, ensuring they were stocked up on feed and had written out care instructions should they no longer be able to visit their horse, despite horse-owning respondents in a similar survey reporting that they were worried that the pandemic would make it difficult for them to meet their horse’s essential needs financially, and that their ability to look after their horse would be affected at some point [4]. Seventeen percent of participants of a US survey on the stressors associated with caring for a pet during COVID-19 reported concerns or difficulties regarding the acquisition of supplies for their pets, including horses, highlighting the hoarding behaviour of other owners as a challenge [8]. Only 9% of participants indicated that they had considered what might happen to their pets should they themselves become ill or otherwise unable to care for them [8]. The lack of preparation is concerning given the potential severity of the situation. Unlike many other areas of the globe, the UK rarely faces natural disasters of this magnitude and the absence of emergency planning may reflect this. That said, the last civil crisis in the UK, the Foot and Mouth Disease (FMD) outbreak in 2001, is well within living memory [14,28]. The devastation this epidemic caused throughout the countryside is unlikely to have been forgotten by many equine owners. Although they are not vulnerable to FMD themselves, the movements of equines were restricted during this time as there was a danger that they could spread the virus, consequently the outbreak also directly affected many equestrians [29]. In more recent years, extreme weather events, such as flooding or snow, have considerably challenged the ability of equine owners to access and provide care for their animals. However, these events tend to be highly geographically localised rather than the nationwide crisis seen with the COVID-19 pandemic. Disaster planning and preparation in advance of the event, particularly in relation to actions on behalf of animals and other vulnerable family members, may help mitigate some of the stress associated with these events and should be something owners are encouraged to think about in advance [14,19]. 

The value of animals in providing emotional support during troubling times, including during the pandemic, is well recognised [8,10,19]. Perhaps of equal value is the comfort they provide by giving people who look after them a sense of purpose and routine [10], things that are lacking in a lockdown environment but are important for mental wellbeing [19,30]. This study did not specifically focus on the mental health implications of the pandemic, but the findings demonstrate the disruption it caused to daily equestrian life and established routine. Many respondents were no longer able to visit their horses spontaneously, if at all, and this is likely to have had considerable implications for their mental health [4]. Not only were these respondents denied social and physical support from people outside their household, but they also lost the psychological benefits of spending time with their equine as well as potentially having the additional concern about their ability to provide adequate care for their animal during this time. That said, for equestrians who were still able to spend time with their horses during this period, the benefits extended beyond the emotional support provided by the horse itself, to increasing physical activity and giving them the opportunity to get outside in contact with nature, both of which also provide mental health benefits [5,7,10]. 

A minority of respondents reported that due to lockdown and being furloughed from work, they were able to spend more time with their horse, a benefit also noted by Merkies et al. in their Ontario-based study [13]. The emotional support provided by equines is likely underestimated in comparison to other companion animal species. However, a 2020 survey of UK residents found that 95% of those with horses, ponies and farm animals believed their animals helped them cope emotionally during the pandemic, and 96% reported that their animals kept them fit and active [9]. In comparison, 91% of respondents with dogs believed their dogs helped them to cope, and 89% of those with cats [9]. It should be noted, however, that 18% of respondents with horses, ponies and farm animals felt it would be easier if they did not have an animal during this time, in contrast with 5% of those with dogs and 4% of those with cats, although the reasoning behind this statement was not explored in detail [9]. Equines are highly sensitive to human emotional state (for example see [31,32] and the potential psychological impact of being exposed to anxious or stressed owners during the pandemic should be acknowledged [33].

## 5. Conclusions

This study focused on the response of equine owners during a short but well-defined period of time at the early days of the COVID-19 pandemic in the UK, when the government began to introduce mandatory changes in human behaviour to control virus spread. At this point, people were unaware of the longer-term changes that controlling the virus would necessitate. Equine owners reported changes to their equine-related routines and activities during this time to minimise risks associated with COVID-19 as well as to mitigate the risk of equine-related accidents. However, a large proportion of the sample continued to ride their equine. The findings highlight several areas that would significantly benefit from in-depth investigation in future research. Equestrian behaviour and mindset around risk-taking and risk perception have already been researched in relation to equestrian activities and sport but have received little attention in the context of wider health challenges. Understanding the uptake of emergency planning and preparation in the UK equestrian community also warrants consideration. Using this information effectively to promote forward planning is likely to be of great benefit in equestrian responses to future health or climate-related crises. 

## Figures and Tables

**Figure 1 animals-11-01215-f001:**
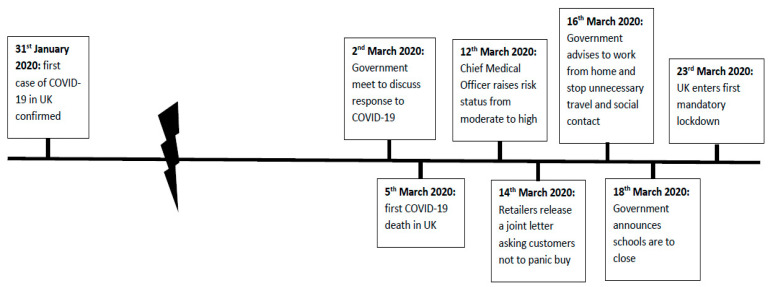
Timeline of key events in the UK January to March 2020 relating to COVID-19.

**Table 1 animals-11-01215-t001:** Demographic information on the survey respondents.

Variable	Answer Options	*n* (%)
Age	18–25 years	45 (10.0%)
	26–40 years	141 (31.2%)
	41–50 years	107 (23.7%)
	51–65 years	138 (30.5%)
	66–75 years	21 (4.6%)
	76+	0 (0.0%)
Country	England	331 (73.2%)
	Scotland	40 (8.8%)
	Wales	31 (6.9%)
	Northern Ireland	1 (0.2%)
	Isle of Man/Jersey	2 (0.4%)
	UK unspecified	47 (10.4%)
High COVID risk status	Yes	101 (22.3%)
	No	331 (73.2%)
	Unsure	19 (4.2%)
	Rather not answer	1 (0.2%)

**Table 2 animals-11-01215-t002:** Reported changes in equine-related activities following the COVID-19 restrictions announced on 16 March 2020 and the closure of schools on 18 March 2020.

Variable	Answer Options	16 March	18 March
		*n* (%)	*n* (%)
Response to restrictions	Carried on normal activities	313 (69.2%)	166 (36.7%)
	Changed my behaviour	138 (30.5%)	167 (36.9%)
	Not sure	1 (0.2%)	-
	Kept to previous changes	-	119 (26.3%)
Stopped riding/changed training	Yes	73 (16.2%)	102 (22.6%)
	Yes stopped some interactions	67 (14.8%)	93 (20.6%)
	No	289 (63.9%)	143 (31.6%)
	Other	23 (5.1%)	-
	Kept to previous changes	-	114 (25.2%)

**Table 3 animals-11-01215-t003:** Reported changes in respondent behaviour following the COVID-19 restrictions announced on 16 March 2020 and the closure of schools on 18 March 2020, categorised from the responses left in the free-text boxes.

Category	16 March (*n* = 150)	18 March (*n* = 224)
	*n* (%)	*n* (%)
Reduced time at yard/number of visits	31 (20.7%)	53 (23.7%)
Followed biosecurity measures at yard	42 (28.0%)	28 (12.5%)
Stopped riding	46 (30.7%)	58 (25.9%)
Took steps to minimise risk (riding or handling)	29 (19.3%)	67 (29.9%)
Changed their equine’s management	8 (5.3%)	15 (6.7%)
Stocked up on feed	2 (1.3%)	5 (2.2%)
Cancelled planned/regular equine activities	13 (8.7%)	20 (8.9%)
Non-equine-related changes	14 (9.3%)	10 (4.5%)
Self-isolating	4 (2.7%)	5 (2.2%)

## Data Availability

Not applicable.

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
