# Peer review of "Response of UK Horse, Pony and Donkey Owners to the Early Stages of the COVID-19 Pandemic"

_animals, 2021, doi:10.3390/ani11051215_

Round 1

Reviewer 1 Report

In my opinion manuscript No animals-1180623 is interesting, well and clearly prepared, readable and it contributes new findings to the knowledge on the human-animal relationships’ during pandemy of Covid-19. The revised manuscript reports original and new data. A cross-sectional studies done in UK showed that following the government suggestions and restriction was challenging for UK equestrians who had to adjust their equine’s routine and daily care alongside the increasing biosecurity measures. Data were collected from 452 respondents via social media to try to understand equine-related decision making at the first lockdown by the equids’ owners. The conclusions withdrawn based on this cross-sectional studies can promote forward planning in case future health or climate related crises.

I have some minor comments.

Specific comments:

L14 space

L57 underlining should be removed

L73 bracket

L116 space

L127 semicolon should be removed, bracket should be added

L299 space

Figure 1 should be: 23rd March 2020 UK enters first mandatory lockdown.

Author Response

Thank you for your kind feedback on our paper. We have responded to your comments as detailed below.

Response to Reviewer 1:

L14 space

Done

L57 underlining should be removed

Done

L73 bracket

Done

L116 space

Done

L127 semicolon should be removed, bracket should be added

Done

L299 space

Done

Figure 1 should be: 23rd March 2020 UK enters first mandatory lockdown.

This has been changed

Reviewer 2 Report

see attachment

Author Response

Thank you for your kind feedback on our paper. We have responded to your comments as detailed below.

55- 57 The paragraph begins with a mention of financial challenge, but the paper doesn’t not follow this  up – the questionnaire didn’t address it, from the looks of it. Given that a lot has been made of the  financial stress – impact on Riding Schools and that side of the industry – it would be good to clarify that  this research doesn’t investigate this, but rather focuses upon understanding the mind sets and  subsequent behaviour of respondents instead.

Good point. We have added the following to acknowledge this: The impact of the pandemic on the practicalities of providing care for horses, donkeys and mules in the UK forms the focus of this study.

90 The phrase “provided participant information” is unclear. Should it be ‘provided information to the participants’?

This has been changed

96 “Country” presumably should be ‘County’

No this was meant to be country (so England, Wales, Scotland, Northern Ireland)

197 – In “Figure 1. Timeline of key events” the last box on the top of the line has a date which needs checking.

This has been changed

201 Mention is made of a “Whole Welfare Framework” but it is not explained here or elsewhere. This  needs to be opened up and explained, or removed.

We have amended this paragraph to try to open up our thinking here: The COVID-19 pandemic is being increasingly viewed through the lens of the One Welfare framework, given the impact it has had on human, animal and environment alike, and how it has highlighted the interconnectedness between them [17,18]. The changes experienced for equines and their owners during this time represent only a small part of this whole but demonstrate how a global health challenge faced by the human population can significantly impact the lives, and potentially the welfare, of one type of domestic animal at a local level.

225 – 240 This paragraph is unclear. It discusses the older cohorts’ responses in the survey. Speculating  on the reasons for their reporting it says: “More difficult to explain is that more high-risk respondents  had previously received hospital treatment for an equine-related accident.” It then goes on to speculate  about potential causes such as ‘health-related paranoia’, before going on to suggest that maybe it is  because they have been riding longer. This is a bit unsatisfying – surely those who have rode longest  would be more likely to report more ‘equine-related accidents’ – which are quite different from human[1]related infection. It would be useful to revisit this paragraph.

We take your point and have amended the paragraph to clarify: More difficult to explain is that more high-risk respondents had previously received hospital treatment for an equine-related accident. This could potentially reflect the compromised health status of some members of this sample or health-related paranoia. However, it is more likely that the higher median age of this sub-sample means that they have potentially been around equines for longer and may therefore be expected to have accumulated more equine-related injuries than younger respondents (who are typically at lower risk of contracting COVID-19). The culture of UK equestrianism where taking risks is normalised may also contribute to this finding [21].

337 A sentence ends with “a benefit also noted by [13].” It would be better to actually state who it is  who noted it, rather than forcing the reader to look to the reference list

We have revised this sentence accordingly: A minority of respondents reported that due to lockdown and being furloughed from work, they were able to spend more time with their horse, a benefit also noted by Merkies et al. in their Ontario based study [13].

Reviewer 3 Report

Thank you for your interesting and current paper submission. I just have a few very minor comments for clarity, but otherwise this was a well-considered paper, thank you.

Line 19 – it’s a bit picky but I’m not a fan of the word “equines” how about “horses, donkeys or mules” in the simple summary instead? You use “equines” or “their equine” repeatedly and I get why you have used this as you are talking about all possible Equidae, so please ignore this comment if you are happy with your use of “equines” to encompass a range of species.

Line 37 – 40 – You mention in the simple summary that owners did not seem to have plans in place for care of their animals if they were ill, but I don’t think you made this point as clearly in the abstract and I feel it is a really important point to include.

Line 73 – at the start of the line you either have too many brackets in place or not enough but the ( are not closed

Line 102 – second sentence doesn’t quite make sense, who were the guidelines suggested by?

Line 263 – not sure that this sentence makes sense i.e. together? Do you mean with other people or did you mean “altogether”?

Line 277 – not sure that this sentence makes sense

Line 283 – should this “given” be “give”

Author Response

Thank you for your kind feedback on our paper. We have responded to your comments as detailed below.

Line 19 – it’s a bit picky but I’m not a fan of the word “equines” how about “horses, donkeys or mules” in the simple summary instead? You use “equines” or “their equine” repeatedly and I get why you have used this as you are talking about all possible Equidae, so please ignore this comment if you are happy with your use of “equines” to encompass a range of species.

We totally understand your preference and have changed the summary wording accordingly

Line 37 – 40 – You mention in the simple summary that owners did not seem to have plans in place for care of their animals if they were ill, but I don’t think you made this point as clearly in the abstract and I feel it is a really important point to include.

Good point. We have added the following to the abstract: Few respondents reported having an emergency plan in place should they become ill or otherwise unable to care for their equine.

Line 73 – at the start of the line you either have too many brackets in place or not enough but the ( are not closed

This has been corrected

Line 102 – second sentence doesn’t quite make sense, who were the guidelines suggested by?

We have reworded this sentence to clarify what we meant: These were (i) the restrictions suggested by the UK government on 16th March 2020 to stop unnecessary travel and social contact, and to work from home if possible – these were suggestions rather than mandatory; and (ii) following the closure of schools on 18th March 2020 [2].

Line 263 – not sure that this sentence makes sense i.e. together? Do you mean with other people or did you mean “altogether”?

We meant altogether – this has been changed

Line 277 – not sure that this sentence makes sense

We have reworded this sentence to clarify what we meant: Data from a South Australian hospital demonstrated a reduction in equestrian-related traumatic injuries received in during 2020 when COVID-19 restrictions were in place, when compared with the same period in 2019, suggesting that Australian equestrians there had also taken measures to reduce their risk of injury [26].

Line 283 – should this “given” be “give”

Yes - changed